# Variation in health visiting contacts for children in England: cross-sectional analysis of the 2–2½ year review using administrative data (Community Services Dataset, CSDS)

Caroline Fraser,[1] Katie Harron,[1] Jane Barlow,[2] Samantha Bennett,[3] Geoffrey Woods,[4] Jenny Shand,[5,6] Sally Kendall ,[7] Jenny Woodman  [8]

For numbered affiliations see end of article.

**Correspondence to**
Jenny Woodman;
j.woodman@ucl.ac.uk

## ABSTRACT

**Objective** The 2–2½ year universal health visiting review in England is a key time point for assessing child development and promoting school readiness. We aimed to ascertain which children were least likely to receive their 2–2½ year review and whether there were additional non-mandated contacts for children who missed this review.

**Design, setting, participants** Cross-sectional analysis of the 2–2½ year review and additional health visiting contacts for 181 130 children aged 2 in England 2018/2019, stratified by ethnicity, deprivation, safeguarding vulnerability indicator and Looked After Child status.

**Analysis** We used data from 33 local authorities submitting highly complete data on health visiting contacts to the Community Services Dataset. We calculated the percentage of children with a recorded 2–2½ year review and/or any additional health visiting contacts and average number of contacts, by child characteristic.

**Results** The most deprived children were slightly less likely to receive a 2–2½ year review than the least deprived children (72% vs 78%) and Looked After Children much less likely, compared with other children (44% vs 69%). When all additional contacts were included, the pattern was reversed (deprivation) or disappeared (Looked After children). A substantial proportion of all children (24%), children with a 'safeguarding vulnerability' (22%) and Looked After children (29%) did not have a record of either a 2–2½ year review or any other face-to-face contact in the year.

**Conclusions** A substantial minority of children aged 2 with known vulnerabilities did not see the health visiting team at all in the year. Some higher need children (eg, deprived and Looked After) appeared to be seeing the health visiting team but not receiving their mandated health review. Further work is needed to establish the reasons for this, and potential solutions. There is an urgent need to improve the quality of national health visiting data.

## Strengths and limitations of this study

► This is the first study to analyse the coverage and intensity of health visiting in England, also taking into account additional (non-mandated) contacts from the health visiting team.

► We addressed incompleteness in the national administrative data on health visiting in England by limiting our analyses to subsets of most complete data by (1) developing methods to identify a research-ready subset of the national data using comparisons to reference data sources and (2) limiting analyses to local areas with <10% missing data for vulnerability indicators.

► Our approach to dealing with incomplete data (including only most complete data) limits the generalisability of our results to the whole of England, with particular implications for results about vulnerable children and different ethnic groups.

► We were reliant on the information recorded in the administrative data, all of which is entered by the health visiting teams.

► Despite the limitations, this is an important contribution to the evidence-base about how health visiting is delivered in England, which is foundational to making any improvements or modifications to the service.

## INTRODUCTION

In England, there has been a sustained cross-government focus on identifying services and policies for babies and young children to reduce inequalities.[1–5] These policies represent a response to the evidence that at age 5 year, certain groups of children are so far behind in terms of development that they will 'struggle to ever catch up'.[6] Children who start school with lower than expected levels of development are more likely to be excluded from school or have social services involvement by the time they are 11.[6] By age 16,

BMJ

disadvantaged children are 18 months behind their peers and 40% of this development gap had already emerged by the age of 5.[6]

There is one universal intervention programme for preschool children in England, which specifically aims to address inequalities and promote health and development of young children: The Healthy Child Programme (HCP 0–5, see figure 1), led by Health visitors. Health visitors are specialist child and family public health nurses who lead a team of mixed skill staff (see figure 1). There are models of health visiting in some countries of the world: Child Health Nurses (Sweden), Public Health Nurses (America Canada, Ireland), Child and Family Health Nurses (Australia), Plunket Nurses (New Zealand), Social Nurses (Belgium), Lady Health Visitors and Lady Health Workers (Pakistan) and Patronage Nurses (Serbia, Kosovo, Kazakhstan). However, health Visiting in England is unique in its universal coverage (figure 1). A key part of the HCP 0–5 is five mandated contacts by the health visiting team before a child turns 3 year old (figure 1). The last of these reviews is at 2–2½ year, and represents a vital opportunity to assess a child's readiness to learn, their physical and social and emotional development and to identify any additional support needed to start school on a level footing with their peers.[6 7] Since 2018, there has been particularly high cross-government policy focus on the role that the 2–2½ year review can play in reducing social inequalities in cognitive development.[8–10] As part of a holistic assessment of the child during the 2–2½ year review, practitioners are required to use the Ages and Stages Questionnaire (ASQ-3[11]), and since 2020, the Early Language Identification Measure (ELIM) is recommended. See figure 1 for a more detailed explanation of these tools.

There is now extensive recognition that the critical 1001 days (conception to age 2) represents the best opportunity for intervening with the aim of reducing inequalities, and the Leadsom report (March 2021) highlights the need for every local authority (LA) to develop a Start for Life offer, in order to achieve this.[10] To evaluate the impact of specific policies and interventions delivered in the critical 1001 days, we need a measure of child development for all children at the end of this period (age 2) which is complete, accurate and available for analysis, such as the ASQ-3 and/or ELIM.

Despite recognition of the importance of the 2–2½ year review both for supporting individual families and for collecting data to support planning and policies at a local and national population level, there is evidence that a substantial proportion of children still do not receive a 2–2½ year review. Data from Public Health England (PHE) 'interim reporting metrics' (referred to as 'metrics' in this paper) indicate that 22% of eligible children in England did not have a record of 2–2½ year review in 2018–2020, with substantial variation across the country (27%–97%).[6 12] PHE has now been replaced with Office for Health Improvement and Disparities (2021) so we refer to (ex)PHE. The Children's Commissioner and

(ex)PHE both estimate that 9 out of 10 children receiving a 2–2½ year review also had an ASQ-3 completed but other studies have found significant variability across England in the implementation and reporting of the ASQ-3.[13–15]

The subuniversal reach of the 2–2½ year review has prompted questions about whether some children are systematically more likely to miss out on a review. In 2020, the Children's Commissioner tried to ascertain if vulnerable children were differentially likely to have a 2–2½ year review than their peers but found that most LAs did not collect the necessary data, concluding: 'There is little evidence that local areas are ensuring that their vulnerable young children are checked'.[13] Since then, (ex)PHE has published experimental statistics,[16] suggesting that the likelihood of receiving mandated reviews varies with ethnicity, and that children living in the most affluent areas of England were more likely to receive a mandated review than children living in the more deprived areas.[10] However, these emerging social patterns need further confirmation. Additionally, we do not know whether additional (non-mandated) contacts by the health visiting team are similarly patterned. It may be, for example, that the children who miss out on a mandated review are seeing members of the health visiting team regularly for other reasons. More recent experimental statistics from (ex)PHE (June 2021) suggested that children in the most deprived neighbourhoods are more likely to get additional contacts from the health visiting, but these analyses did not also include the mandated reviews.[17 18]

## AIM

We aimed to ascertain whether certain groups of children were less likely to receive their 2–2½ year review than other children. We used a national administrative dataset (the Community Services Dataset; CSDS)[19] to calculate the percentage of children in 2018/2019 who received their 2–2½ year review, stratified by ethnic group, deprivation quintile, safeguarding vulnerability and Looked After Child Status. We investigated whether those that missed out on their 2–2½ year review were seeing the health visiting team for other reasons in the same time period.

## METHODS
### Study design and setting

This study comprises a cross-sectional analysis of coverage of the 2–2½ year health visiting review and coverage and intensity of additional health visiting contacts for children aged 2 in England in 2018/2019, stratified by ethnic group, deprivation and 'vulnerability' indicators.

### Data source

The CSDS contains individual-level longitudinal administrative data from community services in England since 2015, including data on mandated and additional contacts with health visiting services. It is operated by National

**Delivery of the Health Child Programme (HCP) by Health Visitors**

The Healthy Child Programme (HCP) is designed to promote the health and development of young children, and prevent and mitigate the impact of adversity and inequalities in early childhood.(1, 2). The HCP is led by health visitors: registered nurses or midwives who have undertaken an additional Specialist Community Public Health Nursing qualification. Health visiting teams comprise health visitors with a specialist qualification in public health and other clinical skill mix staff such as band 5 staff nurses and nursery nurses. There is wide variation in the skill mix within health visiting teams: in February 2020 health visitors made up on average 70% of teams, but there was substantial variation across local areas, with a range from 33% to 100%.(3) Studies report that a minority (between a third (4) and a fifth (31)) of 2-2½ year reviews are conducted by qualified health visitors in England and that 21% of heath visitors report that families in their local area always received this review from a qualified health visitor (5). As part of the HCP, LAs are required to commission delivery of a minimum of five contacts for every child and family in England before the child is aged 5 years in which health visitors or other members of the team review the child's health and development in the context of family health and environment, offer support in a range of areas, and signpost to community resources such as children's centres, child care and wider early years services.(6) The health visiting contacts can be delivered through home visits, individual or group clinic appointments, or phone calls and are supported by administrative activity such as letters to families.(7) The five mandated reviews, also known as 'universal health reviews', are an antenatal visit after 28 weeks of pregnancy, a new birth visit within 14 days, a 6 to 8 week review, an age one year review and a 2-2½ year review.(1)

**Ages and Stage Questionnaire (ASQ-3)**

The ASQ-3 is a tool to measure child development. Policy guidance for England states that the ASQ-3 should always be used as part of the 2-2½ year review (8) and practitioners might also choose to use the ASQ-SE™, social and emotional questionnaire.(9) In 2020 the Office for Children's Commissioner in England estimated that 9 in 10 children with a 2-2½ review had a ASQ-3 assessment. (4) ASQ as a screening tool for developmental delay in children <5y delivered by Paediatricians has been compared to other screening tools in American populations and judged to have "modest sensitivity" for detecting developmental delay but "adequate specificity" (i.e. they missed lots of children with developmental delay but didn't pick up too many false positives). (10) There is no equivalent study for ASQ-3 as delivered by health visitors and nursery nurses in an English setting, though an analysis done by Ofsted and NHS Digital (2017) reported that scores on ASQ-3 used at the 2-2½ year review were not well correlated at either a national or local level with the Early Years Foundation Stage Profile (an developmental assessment routinely completed for every child by their teacher at the end of the first year of school).(11) The lack of studies validating ASQ-3 as an individual level screening tool for developmental delay in an English setting drives the English policy guidance that ASQ-3 should be used as a *population measure* of child development for monitoring and to inform interventions, not as an individual assessment or screening tool. (8) However, a study in 4 sites in England (2014) found confusion among parents and health care professionals about the purpose of ASQ-3, namely whether it was a tool for assessing a child's individual development or a population measure of child development(12). In 2020 the Office for the Children's Commissioner for England found evidence that ASQ-3 was being used in a large number of local authorities as a screening tool.(4) The 2014 study in 4 sites in England concluded that variation in the use of ASQ (e.g. sometimes completed by professionals and sometimes by parents) potentially undermines its usefulness as a population measure of child development (13)

**Early Language Identification Measure (ELIM)**

ELIM-E is a tool designed for use in the 2-2½ year review in order to identify children with speech, communication or learning needs and which should be used as part of a local pathway to identify and support individual children towards school readiness and, ultimately, reduce inequalities in school readiness across the population.(14, 15) The Early Language Identification Measure (ELIM) and accompanying intervention was commissioned in 2018 as part of the UK government's Social Mobility Action Plan,(16) with guidance issued to commissioners, service leads and practitioners in late 2020.(15, 17) Unlike the ASQ-3 tool, Public Health England and Department of Health and Social Care recommend the use of ELIM-E for assessing need in individual children.

**Figure 1** Delivery of the HCP by health visiting teams and use of ASQ-3 and ELIM in the 2–2½ year review.[15 29–44]

Health Service (NHS) Digital, and providers of publicly funded community services (including NHS trusts, private providers and the voluntary sector) are legally mandated to submit data. CSDS captures basic child characteristics (age, ethnicity), contacts with health visiting services (type, frequency, length, date) and a wider range of identified needs in children such as referral from or to specialist services. Quintiles of deprivation are derived from the Index of Multiple Deprivation (IMD), based on the child's postcode of usual address.

CSDS is the only child-level national source of information about health visiting in England but is a relatively new dataset, with outputs classified by NHS Digital as experimental. To our knowledge, the only published analysis of CSDS has been (ex)PHE's experimental statistics describing patterns in mandated reviews, cited by the Leadsom review[10 16] and (separately) patterns of additional contacts.[17]

### Our 'research-ready' subset of data

We analysed a pseudonymised extract of CSDS that was held within (ex)PHE for the purpose of delivering (ex)PHE's core work programme and priorities, for contacts between 1 April 2018 and 31 March 2019. To account for variation in the completeness of CSDS data across England and over time, we developed and applied methods for identifying a subset of 'research-ready' CSDS data for analysis (full details in online supplemental material 1). The research-ready subset of data was restricted to LAs with a high level of data completeness. We checked the completeness of CSDS by comparing the number of eligible children and health visitor contacts recorded in CSDS with ONS data on births, health visitor contacts reported within (ex)PHE metrics and anonymised health visiting data obtained directly from three LAs.

Our research-ready dataset for analyses of the 2–2½ year review in 2018/2019 included 181 130 children from 33 LAs. Children in the research-ready dataset were similarly deprived but less ethnically diverse than all children in England (see online supplemental material 1). We identified a further subset of 13 LAs with 18 240 children with sufficiently complete data (<10% missing) for analyses of Looked After Child Status and 7 LAs with 15 485 children with sufficiently complete data on the data item 'Safeguarding Vulnerability'.

### Analyses

We estimated the percentage of eligible children who received a 2–2½ year review. Our denominator comprised all children aged 2 on 31 March 2019 minus the number of children who had the 2–2½ year review and were aged 3 on 31 March 2019 (see online supplemental material 2) for full details). The numerator comprised children aged 2 on 31 March 2019 who had a 2–2½ year review scheduled and coded as 'attended' or with missing attendance data (5% of all 2–2½ year reviews).

We calculated the percentage of children who received any face-to-face contact with a member of the heath visiting team for any reason (including the mandated 2–2½ year review and additional contacts) in 2018/2019, by location of contact (home/any other location). We calculated the median and IQR for the number of all attended contacts per child (including letters and phone calls) and face-to-face 'attended' contacts. We quantified the number of children with a recorded ASQ-3 but no 2–2½ year review recorded, as an indication of 2–2½ year reviews that took place but might not have a 2–2½ year review code attached.

We stratified results by ethnicity, deprivation (IMD quintile) and vulnerability, where a 'vulnerable' child was defined as one with a code indicating a safeguarding vulnerability or Looked After Child at any point during 2018/2019. Both codes are entered manually by a member of the health visiting team as part of their usual care of the child. The 'safeguarding vulnerability' code indicates factors such as referral from police or children's social care, significant injury, known or suspected domestic abuse, worrying parent behaviour or concerns about parental mental health.[19] Analysis of these characteristics was only conducted for LAs with less than 10% missing data for these variables (N=13 for Looked After Children and N=7 for safeguarding vulnerability). As IMD was complete for all 33 LAs and ethnicity had a similar level of completeness (16%–23% missing) across all LAs, we included 33 LAs in analyses of these variables.

### Patient and public involvement

We did not conduct any patient or public involvement for this study.

### RESULTS

In our research-ready dataset, 74% of eligible children received their 2–2½ year review, 76% had any face-to-face contact with the health visiting team in the previous 12 months (including a 2–2½ year review and additional contacts) and 78% had any contact, including letters or phone calls (table 1). If we assume that all the children with an ASQ-3 record had a 2–2½ year review, our estimate of children with a 2–2½ year review increases to 81%.

The most deprived quintile of children were less likely to have received a 2–2½ year review (72%) than children in the least deprived quintile (78%), with a gradient across quintiles. This pattern was reversed (on a small scale) when additional contacts were taken into account (80% for the most deprived vs 78% for the least deprived quintile, table 1). Looked After Children were much less likely to have received a 2–2½ year review recorded compared with other children (44% compared with 69%, table 1). However, this difference disappeared when all face-to-face additional contacts with the health visiting team were included. The lower proportion of 2–2½ year reviews for Looked After Children was not explained by missed appointments: all 90 Looked After Children with scheduled 2–2½ year reviews were recorded as 'attended'.

**Table 1** Percentage of children aged 2 with a 2–2½ year review or any contact (face-to-face or other medium) with a health visiting team

| Child characteristic | With a 2–2½ year review attended | | Children aged 2 on 31 march 2019 with any contact with health visiting teams | | | |
| --- | --- | --- | --- | --- | --- | --- |
| | | | Face-to-face contacts | | Any (including letters and telephone) | |
| | % of eligible* children (95% CI) | No with a review/ eligible children | % of eligible* children (95% CI) | No with a review/ eligible children | % of eligible* children (95% CI) | No with a review/ eligible children |
| All children (33 LAs) | 74% (73% to 74%) | 55 975/75 960 | 76% (75% to 76%) | 70 695/93 525 | 78% (78% to 78%) | 73 065/93 525 |
| Safeguarding vulnerability code (7 LAs) | | | | | | |
| No | 86% (85% to 86%) | 13 090/15 280 | 80% (80% to 81%) | 14 550/18 150 | 83% (83% to 84%) | 15 120/18 150 |
| Yes | 83% (78% to 88%) | 170/205 | 78% (73% to 83%) | 195/250 | 80% (75% to 85%) | 200/250 |
| Looked after child (13 LAs) | | | | | | |
| No | 69% (69% to 70%) | 19 475/28 035 | 77% (77% to 78%) | 26 820/34 705 | 79% (78% to 79%) | 27 355/34 705 |
| Yes | 44% (37% to 51%) | 90/205 | 71% (65% to 77%) | 170/240 | 73% (67% to 79%) | 175/240 |
| Quintile of IMD (33 LAs) | | | | | | |
| 1 (most deprived) | 72% (72% to 73%) | 14 295/19 800 | 78% (78% to 79%) | 18 620/23 785 | 81% (80% to 82%) | 19 305/23 785 |
| 2 | 72% (71% to 72%) | 11 065/15 475 | 76% (76% to 77%) | 14 420/18 910 | 79% (78% to 79%) | 14 870/18 910 |
| 3 | 73% (72% to 74%) | 10 685/14 680 | 74% (73% to 74%) | 13 630/18 465 | 76% (75% to 77%) | 14 035/18 465 |
| 4 | 75% (75% to 76%) | 9850/13 075 | 73% (73% to 74%) | 11 985/16 345 | 76% (75% to 77%) | 12 420/16 345 |
| 5 (least deprived) | 78% (77% to 78%) | 10 045/12 940 | 76% (75% to 76%) | 12 165/16 015 | 78% (78%, 79%) | 12 565/16 015 |
| Ethnicity (33 LAs) | | | | | | |
| White | 73% (72% to 73%) | 33 455/45 920 | 73% (72% to 73%) | 42 360/58 150 | 75% (75% to 75%) | 43 655/58 150 |
| Mixed | 79% (78% to 80%) | 5875/7405 | 90% (89% to 91%) | 7390/8210 | 91% (91% to 92%) | 7505/8210 |
| Asian or Asian British | 76% (75% to 78%) | 2920/3825 | 80% (78% to 81%) | 3555/4470 | 81% (80% to 82%) | 3620/4470 |
| Black or black British | 73% (71% to 75%) | 1185/1620 | 80% (79% to 82%) | 1530/1905 | 82% (80% to 83%) | 1555/1905 |
| Other | 63% (61% to 66%) | 1045/1650 | 71% (69% to 73%) | 1450/2030 | 75% (73% to 77%) | 1520/2030 |

*The percentage of children aged 2 with any face-to-face contact may be lower than the percentage of children with a 2–2½ year review to account for children in the denominator for the 2–2½ year review because we reduced the number of children with a 2–2½ year review aged 3 (based on age at end of 2018/2019), and a small number of 2–2½ year reviews were not be face to face (see online supplemental material 2 for full explanation).
IMD, Index of Multiple Deprivation; LAs, local authorities.

The small number of children with a safeguarding vulnerability (n=205), were slightly less likely to have received a 2–2½ year review (83%) than other children (86%) and this held true when additional contacts were taken into account. Three were no clear patterns of 2–2½ year reviews across ethnic groups (table 1). However, analyses that included additional contacts showed there was a lower proportion of white children receiving contacts (73%) than children in the other ethnic categories (80%–90%, table 1).

The majority of all children (76%), children with safeguarding vulnerabilities recorded (78%) and Looked After children (71%), had a record of a contact with health visiting services, either for a 2–2½ year review or as an additional contact (table 1). The most common face-to-face activities (n=110 780) for children aged 2, other than the 2–2½ year review, were 'other' (31%, n=34 180), assessments (23%, n=26 015), counselling, advice or support (17%, n=18 875) and clinical interventions (15%, n=17 145), which include individualised care plans, for example parenting advice, support with behaviour and child development. These figures include multiple activities recorded for the same child (including on the same day).

A substantial proportion of all children (24%), children with a 'safeguarding vulnerability' recorded (22%) and Looked After children (29%) did not have a record of either a 2–2½ year review or any other face-to-face contact (table 2). Most of the children without a face-to-face contact did not have a record of letters or telephone calls from health visiting services, suggesting no contacts were attempted. On average, children that had any type of contact with health visiting services saw a member of the team once in this period, compared with three times for children with a safeguarding vulnerability code and five times for children with a Looked After Child code (table 2).

Overall, 28% of children with a face-to-face contact were seen at home, and the percentage of at home contacts was greater for Looked After children (63%), children with safeguarding vulnerabilities (78%), and children living in the most deprived areas (38%) (table 3).

However, if all contacts with missing data on location (12%; 8805/70 695) were in fact home visits, the proportion of all face-to-face contacts that were in a child's home could be as high as 40%.

## DISCUSSION
### Main findings

The majority (>70%) of eligible children in our sample, including those defined as vulnerable, were engaged with health visiting services and received their 2–2½ year review. Children in the most deprived areas were slightly more likely to miss out on a 2–2½ year review and Looked After children were much more likely to miss out on this review. However, when all additional contacts were included, the pattern was reversed (for deprivation) or

**Table 2** Median contacts per child for children aged 2 with any recorded contact during 2018/2019

| Child characteristic | Median (IQR) no of contacts aged 2 for children with contacts | |
| --- | --- | --- |
| | Any contact | Face to face |
| All children (33 LAs) | 1 (1–2) | 1 (1–2) |
| Safeguarding factors (7 LAs) | | |
| No | 1 (1–2) | 1 (1–2) |
| Yes | 3 (2–7) | 3 (1–7) |
| Looked after child (13 LAs) | | |
| No | 1 (1–2) | 1 (1–2) |
| Yes | 6 (3–10) | 5 (3–9) |
| Quintile of IMD (33 LAs) | | |
| 1 (most deprived) | 2 (1–4) | 2 (1–3) |
| 2 | 2 (1–3) | 1 (1–3) |
| 3 | 1 (1–3) | 1 (1–2) |
| 4 | 1 (1–2) | 1 (1–2) |
| 5 (least deprived) | 1 (1–2) | 1 (1–2) |
| Ethnicity (33 LAs) | | |
| White | 1 (1–3) | 1 (1–2) |
| Mixed | 1 (1–3) | 1 (1–2) |
| Asian or Asian British | 2 (1–2) | 1 (1–3) |
| Black or black British | 2 (1–4) | 1 (1–3) |
| Other | 1 (1–3) | 1 (1–2) |

IMD, Index of Multiple Deprivation; LAs, local authorities.

disappeared (for Looked After children). This highlights the importance of including additional contacts in analyses when investigating whether the delivery of health visiting in England is socially patterned. We also found that Looked After and the most deprived children who were in contact with the health vsiting sevice according to our data received more intensive health visiting than their peers, with more frequent face-to-face contacts. Although we found no evidence that children with a safeguarding vulnerability recorded were any more or less likely to receive a 2–2½ year review or an additional contact than their peers, we did find that when this group had at least one face-to-face contact, they received more intensive health visiting than average.

Our results suggest that there are groups of high need children (eg, deprived children and Looked After children) who are engaged with the health visiting team but do not receive their mandated 2–2½ year review. We could not ascertain how far children without a 2–2½ year review were having their needs formally assessed through other routes such as a statutory Looked After Child review, as part of a Child Protection Plan or had developmental delay or additional needs identified by the health visiting team prior to the 2–2 ½ year review, as suggested by an analysis by the Children's Commissioner in 2020.[13] The same analysis by the Children's Commissioner reported

**Table 3** Percentage of children aged 2 with a record of a face-to-face contact in any location and at home by child characteristic

| Child characteristic | Children aged 2 with a record of a face-to-face contact* | | | |
| | In any location | | At home | |
| | % (95% CI) | Children with a contact/ children aged 2 | % (95% CI) | Children with a contact/ children aged 2 |
|---|---|---|---|---|
| All children (33 LAs) | 76% (75% to 76%) | 70 695/93 525 | 28% (27% to 28%) | 26 130/93 525 |
| Quintile of IMD (33 LAs) | | | | |
| 1 (most deprived) | 78% (78% to 79%) | 18 620/23 785 | 38% (37% to 38%) | 8920/23 785 |
| 2 | 76% (76% to 77%) | 14 420/18 910 | 28% (27% to 29%) | 5345/18 910 |
| 3 | 74% (73% to 74%) | 13 630/18 465 | 27% (26% to 27%) | 4900/18 465 |
| 4 | 73% (73% to 74%) | 11 985/16 345 | 24% (23% to 24%) | 3845/16 345 |
| 5 (least deprived) | 76% (75% to 76%) | 12 165/16 015 | 20% (19% to 21%) | 3210/16 015 |
| Ethnicity (33 LAs) | | | | |
| White | 73% (72% to 73%) | 42 360/58 150 | 27% (27% to 28%) | 15 985/58 150 |
| Mixed | 90% (89% to 91%) | 7390/8210 | 25% (24% to 26%) | 2020/8210 |
| Asian or Asian British | 80% (78% to 81%) | 3555/4470 | 32% (31% to 33%) | 1435/4470 |
| Black or black British | 80% (79% to 82%) | 1530/1905 | 32% (31% to 33%) | 595/1905 |
| Other | 71% (69% to 73%) | 1450/2030 | 31% (29% to 33%) | 630/2030 |
| Looked after children (13 LAs) | | | | |
| No | 77% (77% to 78%) | 26 820/34 705 | 27% (27% to 28%) | 9500/34 705 |
| Yes | 71% (65% to 77%) | 170/240 | 63% (56% to 67%) | 150/240 |
| Safeguarding factors (7 LAs) | | | | |
| No | 80% (80% to 81%) | 14 550/18 150 | 38% (38% to 39%) | 6420/18 150 |
| Yes | 78% (73% to 83%) | 195/250 | 65% (59% to 70%) | 130/205 |

*Rounded to nearest 5 to reduce risk of disclosure as data are subnational.
IMD, Index of Multiple Deprivation; LAs, local authorities.

that some LAs had stated that the 2–2 ½ year review may not always be appropriate when reviewing children with complex heath needs or development delays.[13] Our findings raise similar questions about whether the 2–2½ year review and/or the ASQ-3 is perceived as appropriate for high need groups in the population, both by professionals and parents/carers.

Our results also suggest that there might be two distinct groups of vulnerable children with regard to the receipt of health visiting. We identified a majority group of vulnerable children who received multiple face-to-face contacts within a 12-month period, often in the child's home. This group appears to be receiving intensive health visiting, consistent with the model of proportionate universalism that underpins health visiting in England (a universal service for all children but with greater support and service provision for children in families with identified needs). The second group comprised a substantial minority of vulnerable children who were not in contact with the health visitor team at all: 22% of children with safeguarding vulnerabilities and 29% of Looked After children did not have a record of either a 2–2½ year review or any other face-to-face additional contact in the year. As we did not find evidence of attempted contacts

with these children in CSDS, there is a real possibility that these vulnerable children might not have heard from or seen a member of the health visiting team within the year.

Our results on the 2–2½ year review are largely consistent with the experimental statistics published by (ex) PHE, with the exception of ethnicity.[16] (Ex)PHE found that children from ethnic minority groups were less likely than white children to receive the mandated reviews, including the 2–2½ year review.[16] The difference is likely due to poor data completeness in CSDS and different analytical approaches to dealing with this incompleteness (see online supplemental material 3). Our results on additional contacts are also largely consistent with the (ex)PHE experimental statistics, which suggest that children likely to have higher needs (eg, deprived children and receiving statutory child protection services) receive more additional contacts than average.[17] Our results build on the (ex)PHE statistics by bringing together mandated and additional contacts in one analysis, which facilitates for the first time estimates of children not seeing the health visiting team for any reason in the 12-month period.

## Implications

Our data indicate that the health visiting service fell short of universal reach for children aged 2 year in England in 2018/2019. Together, our results and the (ex)PHE experimental statistics suggest that 2–2½ year reviews are socially patterned. However, the scale of the difference and whether it exists for different ethnic groups remains uncertain, largely as a result of poor data completeness in CSDS (online supplemental material 3). Poor data remain a significant barrier to understanding how health visiting is currently delivered, and the available national data for England can only generate results which should be treated as hypotheses, with a need for further exploration.

Although the health visiting service seems to be achieving a model of proportionate universalism for some children with known vulnerabilities (with multiple visits a year, often in the child's home), there was also a substantial minority of vulnerable children without any recorded contact from the health visiting team. This is of significant concern, given the likely concentration of health and welfare need in these groups and makes it unlikely that the current format and delivery of the 2–2½ year review is helping to address inequalities in school readiness as intended. The secondary effects of the COVID-19 pandemic, including heightened need and reduced workforce capacity (particularly the number of qualified health visitors), are likely to exacerbate the shortfall both in universal reach and provision of extra intensive services where needed.[20]

## CONCLUSIONS

Since completion of this study, the HCP is being revised to be 'universal in reach and personalised in response'.[21] Our study evidences the gap between this policy ambition and the current reality of health visiting.[22] Further work is needed to establish reasons for low coverage of the 2–2½ year review in Children Looked After and to generate recommendations about the most suitable way of delivering the 2–2½ year review to (1) all children and (2) Looked After Children and other groups with known complex needs or vulnerabilities. Relevant factors may include skill mix (health visitor vs other team member, see figure 1), continuity of care, quality of reviews and referral pathways following identification of need.

In order to produce reliable estimates, researchers require access to high quality national data from health visiting services linked to other datasets (such as primary care, hospital, social care and education) for the whole of England.[23] CSDS is not yet sufficiently complete.[24] NHS Digital and (ex)PHE have been undertaking quality improvement work to address this. The findings of this study suggest that policies and/or funding to accelerate the pace of CSDS quality improvements should be considered at a national level. Further work is needed both on how best to support local practitioners to improve data recording in their systems and on how data transfer from

local systems into CSDS might be improved, given the evidence that some data on health visiting contacts in local systems do not make it into CSDS.[24]

## Limitations

Our data did not allow us to follow a child over time, which might mean we slightly underestimate 2–2½ year reviews as some children may have had their 2–2½ year review just before or after our study period. As administrative data, CSDS codes will not capture everything health visitors identify or deliver. Information might be coded with high accuracy on local systems but 'lost' in the transfer to NHS Digital, for example, with use of slightly different codes. There is some evidence of this, based on our comparison of two local areas with CSDS data.[24] Second, contacts may be miscoded. Although our analyses of ASQ-3 records provide some evidence of this, the consistency of our results with other data[24] suggests that miscoding does not explain the one in five children 'missing' a 2–2½ year review. We know that much abuse and neglect of children will not be coded in administrative data, even if it is suspected by professionals[25 26] which will lead to underestimating differences between vulnerable and other children. Many data items were too poorly completed to use, for example, child disability and staff type (eg, health visitor, nursery nurse). CSDS does not capture quality or meaning of the interaction between health visitors and parents. To obtain a full picture of health visiting activity in England, and the contribution of this service to child health and well-being, all types of LAs need to be investigated using a triangulation of data sources, including CSDS, locally held data, surveys and in-depth qualitative data collected from professionals and families, as a study in Scotland is attempting[27] and as we have funding to do for children living with adverse childhood experiences in England.[28]

Our results cannot be confidently generalised to all children in England. The children in the 'research-ready' data were not nationally representative in terms of ethnicity (online supplemental material 1) and the excluded LAs might be systematically different from those we included which had more complete data. Although our findings should be treated as hypotheses, this study marks an important first step in making use of existing, routinely collected data to understand the coverage and intensity of health visiting services in England.

**Author affiliations**
[1]Population, Policy and Practice, UCL Great Ormond Street Institute of Child Health, London, UK, London, UK
[2]Department of Social Policy and Intervention, University of Oxford, Oxford, UK
[3]Public Health Division, Operations Directorate, Kent Community Health NHS Foundation Trust, Ashford, UK
[4]Early Help Data & Information Team, East Sussex County Council, Lewes, UK
[5]UCL Partners, London, UK
[6]Care City, London, UK
[7]University of Kent, Canterbury, UK
[8]Thomas Coram Research Unit, UCL Social Research Institute, London, UK

**Acknowledgements** We would like to thank individuals in Public Health England for their comments and advice on early drafts, including Wendy Nicholson within Nursing, Maternity and Early years and Kate Thurland and Helen Smith. We thank Professor Ruth Gilbert at UCL Great Ormond Street Institute of Child Health for input into study design, methods and drafts. We are grateful to Care City, Barking and Dagenham, Havering and Redbridge CCG, London Borough of Barking and Dagenham, East Sussex County Council and East Sussex Healthcare Trust and Kent County Council and Community Health NHS Foundation Trust for access to their local data and knowledge and expertise. We thank individuals at the Institute of Health Visiting for comments on this paper and earlier drafts.

**Contributors** JW and KH conceived of the study, were responsible for design of the study and supervised all analyses which were carried out by CF. JW, KH, CF, JB, SB, GW and JS all contributed substantially to the design of the study. CF, KH, JB, SB, GW, JS, SK and JW substantially contributed to interpretation of data. JS, GW and SB provided access to locally held data and gave expert advice on data and the service context. CF and JW drafted the paper and KH, JB, SB, GW, JS and SK substantially contributed to its intellectual content, including in multiple revisions. CF, KH, JB, SB, GW, JS, SK and JW approve the final version of the paper for publication. JW and KH are jointly responsible for the overall content as guarantors.

**Funding** This study was funded by the National Institute for Health Research (NIHR) Policy Research Programme, funder reference: PR-PRU-1217-21301; UCL award code: 177763. The views expressed are those of the author(s) and not necessarily those of the NIHR or the Department of Health and Social Care. This project is delivered as part of (ex)PHE's 2020/2021 Life Course Intelligence business plan for child and maternal health in order to understand CSDS data quality and its suitability for further analysis to understand trends, variation and inequalities in health visiting and outcomes for children in the early years. An honorary contract with (ex)PHE enabled the researcher to work as part of the analytical team and analyse the CSDS for this purpose.This research was supported in part by the NIHR Great Ormond Street Hospital Biomedical Research Centre and the Health Data Research UK (grant No. LOND1), which is funded by the UK Medical Research Council and eight other funders. KH is supported by funding from NIHR (17/99/19).

**Disclaimer** The views expressed are those of the author(s) and not necessarily those of the NIHR, the Department of Health and Social Care or Public Health England.

**Competing interests** None declared.

**Patient consent for publication** Not applicable.

**Ethics approval** This study involves human participants and was approved by UCL Institute of Education Research Ethics Committee on 1. May 2020 (Ref 1333). This is a secondary analysis of non-identifiable administrative health data. It would not be feasible to obtain consent, because (1) we do not have access to names and addresses with which to contact individuals, and obtaining these contact details would require a further disclosure of personal information, and (2) we used data on 181 130 children. It would not be possible to contact this large number of families. However, CSDS does not contain any direct patient identifiers such as names, addresses, NHS numbers or full dates of birth. This means the threat to patient anonymity is very low.

**Provenance and peer review** Not commissioned; externally peer reviewed.

**Data availability statement** Data may be obtained from a third party and are not publicly available. The authors do not have permission for onward sharing of the individual-level data underlying this article. Requests to access the CSDS data can be made to NHS Digital.

**ORCID iDs**
Sally Kendall http://orcid.org/0000-0002-2507-0350
Jenny Woodman http://orcid.org/0000-0002-9403-4177

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
