## [Reviewer comments · BMJ Open]

ARTICLE DETAILS

TITLE (PROVISIONAL)	Variation in health visiting contacts for children in England: cross-sectional analysis of the 2-2½ year review using administrative CSDS data
AUTHORS	Fraser, Caroline; Harron, Katie; Barlow, Jane; Bennett, Samantha; Woods, Geoffrey; Shand, Jenny; Kendall, Sally; Woodman, Jenny

VERSION 1 – REVIEW

REVIEWER	Coombes, Lindsey Oxford Brookes University
REVIEW RETURNED	27-Nov-2021

GENERAL COMMENTS	I thought the authors provided a logical and convincing case for the importance of the study. While there is a certain predictability in finding that administrative data about Health Visiting is incomplete and inconsistent, the way the authors have arrived at this conclusion and dealt with these challenges is novel and should be of interest to health professionals and researchers. I thought the introduction to the study was clear and established the rationale for the study. Slightly controversially, one of the ways the authors contextualised the study was by reference to “levelling up” (p. 5, first sentence). I doubt that there are many people who are not familiar with this term, but I suspect there are less who know what it actually means or whether it will be of any use in tackling health inequalities. I suggest that those who have lived through the “Big society” would tend to want to suspend judgement. I would suggest either omitting the term (it doesn’t contribute much to the policy context) or explain it more fully. I will leave it to the discretion of the authors whether they act on these comments. On p. 6 the authors say “Data from Public Health England (PHE) 'interim reporting metrics' ('metrics'). I wondered what the ('metrics') referred to? In the “ARTICLE SUMMARY: strengths and limitations of this study” (p.3) the authors say “Although we restricted our analyses to local authorities with the most complete data, it is likely that there is still misclassification of children as vulnerable/not due to team members not being aware of everything that is happening in a child's life or professional concerns not meeting thresholds of certainty or severity for using the 'safeguarding vulnerability' code in the child's record” I found this difficult to follow after the forward slash in the sentence. There seems to be a double negative in a statement about professionals incomplete awareness of children’s lives and a separate statement in disjunction with this. The aims (objective) are stated in different places in the paper. In the abstract the authors say
---

	“We aimed to establish which children do not receive their 2-2½ year review and whether they have additional (non-mandated) contacts with the health visiting team” (p.2) On p.6 they say the aim was “...to determine whether children from certain ethnic groups in England, or who had higher levels of deprivation or a safeguarding vulnerability were more likely to miss out on their 2-2½ year review in 2018/9 than their peers” and “We aimed to quantify the proportion of children that had additional contacts from the health visiting team in 2018/9 (coverage), the frequency of these contacts (intensity) and how coverage and intensity of additional contacts varied according to ethnicity, deprivation and whether a child had safeguarding vulnerabilities recorded or was recorded as a Looked After Child” (p.6) There is considerable overlap between these statements, but there are differences as well. I would also add in relation to the objective stated in the abstract (identifying which children do not receive their 2-2½ year review) this seems to conflict with the results section of the abstract (children who do receive their 2-2½ year review). On the whole I thought the key concepts used in the study were well explained. However, I wondered if “Research ready” ‘needed some explanation e.g. processed data which has been fully calibrated, combined and cleaned/annotated. Or maybe the term is self-evident (?) I thought that one of the best aspects of the paper was the identification of the strengths and weaknesses of the data sets and how the sample was selected. The key data collection instruments (ASQ-3 and ELIM) relating to the study were discussed. The paper refers mainly to the ASQ-3 especially in Fig. 1. The following statement in Fig. 1 seemed a bit odd to me: “A study in 4 sites in England (2014) found confusion among parents and health care professionals about the purpose of ASQ-3, namely whether it was a tool for assessing a child's individual development or a population measure of child develop. This study recommends ASQ-3 be used as [as] one tool in the wider assessment of a child's health and development.(14) (p.27) Study (14) says parents and professionals are confused about the ASQ-3, but also recommends its use. There is a superfluous “as” in the final sentence. I thought the ASQ-3 material could be strengthened by reference to some of the technical information about it e.g. validity, reliability. On the whole, I thought the findings, discussion and conclusions sections of the paper were clear and addressed the aims of the study. However, on p. 9 the authors say: “More deprived children were slightly more likely to receive a 2-2½ year review than less deprived children but the same pattern was not evident for additional contacts (Table 1). The proportion of eligible children with a 2-2½ year review did not vary meaningfully according to whether the child had a Safeguarding Vulnerability recorded or by ethnicity (see confidence intervals, Table 1)” (p.9) I thought that the first sentence was quite difficult to understand and - with the second sentence - difficult to verify from Table 1. I wondered why the two sentences were not more like the one that followed: “Looked After Children were much less likely to have a 2-2½ year review recorded compared to other children (44% compared to 69%, Table 1) (p.9)
--	--

VERSION 1 – AUTHOR RESPONSE

RESPONSE TO PEER REVIEW

Thank you for the careful and thoughtful reading of our paper and the helpful comments

1. I thought the authors provided a logical and convincing case for the importance of the study. While there is a certain predictability in finding that administrative data about Health Visiting is incomplete and inconsistent, the way the authors have arrived at this conclusion and dealt with these challenges is novel and should be of interest to health professionals and researchers.

- We agree that our study might accelerate health visiting research using administrative data by providing exemplar methods for dealing with incompleteness and inconsistency in the data and that our conclusions should be of interest to health professionals.

2. I thought the introduction to the study was clear and established the rationale for the study. Slightly controversially, one of the ways the authors contextualised the study was by reference to “levelling up” (p. 5, first sentence). I doubt that there are many people who are not familiar with this term, but I suspect there are less who know what it actually means or whether it will be of any use in tackling health inequalities. I suggest that those who have lived through the “Big society” would tend to want to suspend judgement. I would suggest either omitting the term (it doesn’t contribute much to the policy context) or explain it more fully. I will leave it to the discretion of the authors whether they act on these comments.

- Thank you for highlighting the complexity of the levelling-up agenda, including its history. We have deleted the reference to 'levelling up' in this sentence and rephrased this as reducing inequalities. This opening sentence now reads

" In England, there has been a sustained cross-government focus on identifying services and policies for babies and young children to reduce inequalities."

- We have done the same in the final paragraph of the introduction, which now reads:

“There is now extensive recognition that the critical 1001 days (conception to age 2) represents the best opportunity for intervening with the aim of reducing inequalities, and the Leadsom report (March 2021) highlights the need for every local authority to develop a Best Start for Life offer, in order to achieve this.(10)”

3. On p. 6 the authors say “Data from Public Health England (PHE) 'interim reporting metrics' ('metrics')”. I wondered what the ('metrics') referred to?

- Thank you for highlighting this. We were trying to indicate that we use 'metrics' as a shorthand for 'Public Health England interim reporting metrics' in the paper. We have rewritten the clause in brackets to make this clear:

"Data from Public Health England (PHE) 'interim reporting metrics' (referred to as PHE 'metrics' in this paper) indicate that 22% of eligible children in England did not have a record of 2-2½ year review in 2018-20, with substantial variation across the country (27-97%).(6, 12)

4. In the “ARTICLE SUMMARY: strengths and limitations of this study” (p.3) the authors say “Although we restricted our analyses to local authorities with the most complete data, it is likely that there is still misclassification of children as vulnerable/not due to team members not being aware of everything that is happening in a child's life or professional concerns not meeting thresholds of certainty or severity for using the 'safeguarding vulnerability' code in the child's record” I found this

difficult to follow after the forward slash in the sentence. There seems to be a double negative in a statement about professionals' incomplete awareness of children's lives and a separate statement in disjunction with this.

- We agree this was an unwieldy sentence. We have rewritten it as one simple sentence, which now reads:

"We were reliant on the information recorded in the administrative data, all of which is entered by the health visiting teams."

We elucidate on this point in the Discussion.

5. The aims (objective) are stated in different places in the paper. In the abstract the authors say

"We aimed to establish which children do not receive their 2-2½ year review and whether they have additional (non-mandated) contacts with the health visiting team" (p.2)

On p.6 they say the aim was "...to determine whether children from certain ethnic groups in England, or who had higher levels of deprivation or a safeguarding vulnerability were more likely to miss out on their 2-2½ year review in 2018/9 than their peers"

and

"We aimed to quantify the proportion of children that had additional contacts from the health visiting team in 2018/9 (coverage), the frequency of these contacts (intensity) and how coverage and intensity of additional contacts varied according to ethnicity, deprivation and whether a child had safeguarding vulnerabilities recorded or was recorded as a Looked After Child" (p.6)

There is considerable overlap between these statements, but there are differences as well. I would also add in relation to the objective stated in the abstract (identifying which children do not receive their 2-2½ year review) this seems to conflict with the results section of the abstract (children who do receive their 2-2½ year review).

- Thank you for highlighting the inconsistency. We have rewritten our aim so it is now consistent across the abstract and 'aim' section and consistent with the way we write the results in the abstract where we report who did - rather than who did not- receive the review. Relevant sections now read

Abstract

"We aimed to ascertain which children were least likely to receive their 2-2½ year review and whether there were additional non-mandated contacts for children who missed this review. "

Aim

"We aimed to ascertain whether certain groups of children were less likely to receive their 2-2½ year review than other children. We used a national administrative dataset (the Community Services Dataset; CSDS (17)) to calculate the percentage of children in 2018/9 who received their 2-2½ year review, stratified by ethnic group, deprivation quintile, safeguarding vulnerability and Looked After Child Status. We investigated whether those that missed out on their 2-2½ year review were seeing the health visiting team for other reasons in the same time period."

- We also revisited the results and conclusions of the abstract to make sure they addressed the questions laid out in our aims, in the same order as we state our aims. These now read:

"Results

The most deprived children were slightly less likely to receive a 2-2½ year review than the least deprived children (72% v 78%) and Looked After Children much less likely, compared to other children (44% v 69%). When all additional contacts were included, the pattern was reversed

(deprivation) or disappeared (Looked After children). A substantial proportion of all children (24%), children with a 'safeguarding vulnerability' (22%) and Looked After children (29%) did not have a record of either a 2-2½ year review or any other face-to-face contact in the year.

Conclusions

A substantial minority of children aged 2 with known vulnerabilities did not see the health visiting team at all in the year. Some higher need children (e.g. deprived and Looked After) appeared to be seeing the health visiting team but not receiving their mandated health review. Further work is needed to establish the reasons for this, and potential solutions. There is an urgent need to improve the quality of national health visiting data."

6. On the whole I thought the key concepts used in the study were well explained. However, I wondered if "Research ready" 'needed some explanation e.g. processed data which has been fully calibrated, combined and cleaned/annotated. Or maybe the term is self-evident (?) I thought that one of the best aspects of the paper was the identification of the strengths and weaknesses of the data sets and how the sample was selected.

- We have a paragraph on page 6/7 defining what we mean by research-ready data, with signposting to methods in the supplementary materials. We have slightly edited this to be clearer. It now reads:

"We analysed a pseudonymised extract of CSDS that was held within (ex)PHE for the purpose of delivering (ex)PHE's core work programme and priorities, for contacts between 1 April 2018 and 31 March 2019. To account for variation in the completeness of CSDS data across England and over time, we developed and applied methods for identifying a subset of 'research-ready' CSDS data for analysis (full details in Supplementary Material 1). The research-ready subset of data was restricted to LASs with a high level of data completeness.

We checked the completeness of CSDS by comparing the number of eligible children and health visitor contacts recorded in CSDS with ONS data on births, health visitor contacts reported within (ex)PHE metrics, and anonymised health visiting data obtained directly from three LAs."

- We have removed all reference to the 'research-ready' dataset in the abstract as there too few words available to describe what we mean by this term. Instead we refer to 'highly complete' data (which is how we derived our research-ready data). This section of the abstract now reads:

"We used data from 33 local authorities submitting highly complete data on health visiting contacts to a 'research-ready' subset of the Community Services Dataset (CSDS))."

- We now also signal our approach to dealing with the incomplete administrative date in the summary, based on comment 1) above suggesting that this is a key strength of the study and will be of wide interest. This now reads:

"We addressed incompleteness in the national administrative data on health visiting in England (CSDS) by limiting our analyses to subsets of most complete data by a) developing methods to identify a research-ready subset of the national data using comparisons to reference data sources b) limiting analyses to local areas with <10% missing data for vulnerability indicators."

7. The key data collection instruments (ASQ-3 and ELIM) relating to the study were discussed. The paper refers mainly to the ASQ-3 especially in Fig. 1. The following statement in Fig. 1 seemed a bit odd to me: "A study in 4 sites in England (2014) found confusion among parents and health care professionals about the purpose of ASQ-3, namely whether it was a tool for assessing a child's individual development or a population measure of child develop. This study recommends ASQ-3 be

used as [as] one tool in the wider assessment of a child's health and development.(14) (p.27). Study (14) says parents and professionals are confused about the ASQ-3, but also recommends its use. There is a superfluous "as" in the final sentence. I thought the ASQ-3 material could be strengthened by reference to some of the technical information about it e.g. validity, reliability.

- We have extended the section in Fig 1 on ASQ-3 to include the evidence-base about the validity and implementation of ASQ-3. It now reads:
"The ASQ-3 is a tool to measure child development. Policy guidance for England states that the ASQ-3 should always be used as part of the 2-2½ year review (8) and practitioners might also choose to use the ASQ-SE□, social and emotional questionnaire.(9) In 2020 the Office for Children's Commissioner in England estimated that 9 in 10 children with a 2-2½ review had a ASQ-3 assessment. (4) in order that the ASQ-3 domain scores can be collected and used as population measure of child development, including at a local level to measure and then put in place interventions to address inequalities in child development "between lowest and highest IMD areas".(8) The use of ASQ as a screening tool for developmental delay in children <5y delivered by Paediatricians has been investigated and compared to other screening tools in American populations and judged to have "modest sensitivity" for detecting developmental delay but "adequate specificity" (i.e. they missed lots of children with developmental delay but didn't pick up too many false positives). (10) There is no equivalent study for ASQ-3 as delivered by health visitors and nursery nurses in an English setting, though an analysis done by Ofsted and NHS Digital (2017) reported that scores on ASQ-3 used at the 2-2½ year review were not well correlated at either a national or local level with the Early Years Foundation Stage Profile (an developmental assessment routinely completed for every child by their teacher at the end of the first year of school).(11)

The lack of studies validating ASQ-3 as an individual level screening tool for developmental delay in an English setting drives the English policy guidance that ASQ-3 should be used as a population measure of child development for monitoring and to inform interventions, not as an individual assessment or screening tool. (8,9) However, Aa study in 4 sites in England (2014) found confusion among parents and health care professionals about the purpose of ASQ-3, namely whether it was a tool for assessing a child's individual development or a population measure of child development.(12) and in 2020 the Office for the Children's Commissioner for England found evidence that ASQ-3 was still being used in a large number of local authorities as a screening tool.(4) The 2014 study in 4 sites in England concluded that the variation in the use of ASQ (e.g. sometimes completed by professionals and sometimes by parents) potentially undermines its usefulness as a population measure of child development.(13)

8. On the whole, I thought the findings, discussion and conclusions sections of the paper were clear and addressed the aims of the study. However, on p. 9 the authors say:
"More deprived children were slightly more likely to receive a 2-2½ year review than less deprived children but the same pattern was not evident for additional contacts (Table 1). The proportion of eligible children with a 2-2½ year review did not vary meaningfully according to whether the child had a Safeguarding Vulnerability recorded or by ethnicity (see confidence intervals, Table 1)" (p.9). I thought that the first sentence was quite difficult to understand and - with the second sentence - difficult to verify from Table 1. I wondered why the two sentences were not more like the one that followed: "Looked After Children were much less likely to have a 2-2½ year review recorded compared to other children (44% compared to 69%, Table 1) (p.9)

- We have rewritten this section to reproduce the sentence structure that the reviewer thinks is clearest and also so that our findings about ethnicity and safeguarding vulnerabilities can be verified from Table 1. It now reads

"RESULTS

In our research-ready dataset, 74% of eligible children received their 2-2½ year review, 76% had any face-to-face contact with the health visiting team in the previous 12 months (including a 2-2½ year review and additional contacts) and 78% had any contact, including letters or phone calls (Table 1). If we assume that all the children with an ASQ-3 record had a 2-2½ year review, our estimate of children with a 2-2½ year review increases to 81%.

The most deprived quintile of children were less likely to have received a 2-2½ year review (72%) than children in the least deprived quintile (78%), with a gradient across quintiles. This pattern was reversed (on a small scale) when additional contacts were taken into account (80% for the most deprived v 78% for the least deprived quintile, Table 1). Looked After Children were much less likely to have received a 2-2½ year review recorded compared to other children (44% compared to 69%, Table 1). However, this difference disappeared when all face-to-face additional contacts with the health visiting team were included. The lower proportion of 2-2½ year reviews for Looked After Children was not explained by missed appointments: all 90 Looked After Children with scheduled 2-2½ year reviews were recorded as 'attended'.

The small number of children with a safeguarding vulnerability (n=205), were less likely to have received a 2-2½ year review (83%) than other children (86%) and this held true when additional contacts were taken into account. There were no clear patterns of 2-2½ year reviews across ethnic groups (Table 1). However, analyses that included additional contacts showed there was a lower proportion of white children receiving contacts (73%) than children in the other ethnic categories (80-90%, Table 1)."

Reviewer: 1

Competing interests of Reviewer: None